# A Highly Sensitive Silicon-Core Quantum Dot Fluorescent Probe for Vomitoxin Detection in Cereals

**DOI:** 10.3390/foods14091545

**Published:** 2025-04-28

**Authors:** Caiwen Dong, Yaqin Li, Xincheng Sun, Xuehao Yang, Tao Wei

**Affiliations:** 1College of Food and Bioengineering, Zhengzhou University of Light and Industry, Zhengzhou 450001, China; biodcw@126.com (C.D.); li15903882170@163.com (Y.L.); biosxc@126.com (X.S.); 13949201346@163.com (X.Y.); 2Food Laboratory of Zhongyuan, Luohe 462300, China

**Keywords:** quantum dots, electrostatic adsorption, fluorescence immunity, deoxynivalenol

## Abstract

Vomitoxin is a member of the monotrichous mycotoxin family with a complex chemical structure and significant biological activity. This toxin has strong immunosuppressive toxic effects and can cause serious damage to human and animal health. In this study, an on-site immune detection method based on an immune SiO_2_@QD fluorescent probe was developed, which realized the rapid and quantitative detection of emetic toxins in grains. Polyethyleneimine (PEI) is a polymer containing a large number of amino groups, and the binding of PEI to the surface of quantum dots can serve to regulate growth and provide functionalized groups. A SiO_2_@QD nanotag with good dispersibility and a high fluorescence intensity was synthesized by combining a PEI interlayer on the surface of SiO_2_ nanospheres. Utilizing the electrostatic adsorption of the amino group in PEI, CdSe/ZnS QDs were self-assembled on the surface of SiO_2_ nanospheres. In the stability test, the SiO_2_@QDs could maintain basically the same fluorescence intensity for 90 consecutive days in the dark at 4 °C, showing a high fluorescence stability. The fluorescence-enhanced QD immune probe was formed by coupling with anti-DON monoclonal antibodies through carbodiimide chemical synthesis. For the detection of spiked wheat flour samples, the immuno-SiO_2_@QD fluorescent probe showed excellent sensitivity and stability, the detection limit reached 0.25 ng/mL, and the average recovery rate was 92.2–101.6%. At the same time, the immuno-SiO_2_@QD fluorescent probe is simple to operate, is capable of rapid responses, and has great potential in the rapid detection of vomitoxins in grains.

## 1. Introduction

Deoxynivalenol (DON), a trichothecene toxin, is a mycotoxin produced by various fungal strains, including Fusarium, Cephalosporium, Lactospora, and Trichoderma [1]. Its chemical structure is complex, characterized as a cyclic ketone compound that contains multiple functional groups, such as hydroxyl and alkenes, and it exhibits significant biological activity [2]. This mold is commonly found in cereals such as wheat, corn, and barley, typically appearing as red or white. It not only readily develops during the growth of these cereals but can also persist during storage. This mold thrives in humid, warm, and cool environments, with a higher likelihood of occurrence when the humidity exceeds 20% and at temperatures ranging between 20 and 30 °C [3]. The presence of deoxynivalenol (DON) directly impacts the growth and development of plants, leading to poor root development, stunted growth, decreased yields, and diminished nutrient content, particularly in terms of protein and minerals. Furthermore, DON adversely affects the plant’s immune system, resulting in decreased resistance to other pathogens [4], which can lead to significant economic losses in food crops. In a survey involving over 3000 samples of raw feed materials and compound feed products across more than 20 provinces and cities in China, the detection rate of deoxynivalenol (DON) was found to be as high as 96.4% to 100%, while the exceedance rate for compound feed was 8.9% [5]. Ingesting plants infected with deoxynivalenol (DON) can lead to acute poisoning symptoms in animals, which may include vomiting, diarrhea, neurological disorders, miscarriage, and stillbirth [6]. Additionally, the long-term consumption of low doses of DON in humans may result in chronic health issues, such as liver and kidney damage, as well as gastrointestinal inflammation [7]. Consequently, countries worldwide impose stringent limits on deoxynivalenol (DON) residues in cereals. According to GB 2761-2017, the ‘National Food Safety Standard Limits of Mycotoxins in Foods’, the maximum allowable limit of DON in cereals and cereal products is set at 1000 μg/kg. In the European Union, the upper limit of DON in food intended for infants has been established at 200 μg/kg [8].

Currently, there are three primary categories into which popular vomitoxin detection techniques fall: physicochemical detection, immunoassays, and biological detection [9]. Commonly employed physicochemical analytical techniques, such as gas chromatography (GC) [10], thin-layer chromatography (TLC) [11], high-performance liquid chromatography (HPLC) [12], gas chromatography–tandem mass spectrometry (GC-MS/MS) [13], and liquid chromatography–tandem mass spectrometry (LC-MS/MS) [14], demonstrate a high sensitivity but generally rely on sophisticated and costly instrumentation. These techniques primarily make use of the unique physicochemical properties of vomitoxin. High sensitivity, objectivity, and accuracy are characteristics of physical and chemical detection techniques, and the results they produce are not influenced by the subjective preferences of the inspector. An immunoassay is a principle-based immunology-based detection method that relies primarily on the specific binding of an antigen and antibody to achieve a desired result. It has a high throughput, sensitivity, and specificity and can detect multiple samples simultaneously to enable target trace analysis [15]. The most common types of immunoassays are the enzyme-linked immunosorbent assay (ELISA) [16], radioimmunoassay (RIA) [17], fluorescence immunoassay (FIA) [18], and chemiluminescence immunoassay (CLIA) [19], all of which are very demanding of the operator and the environment in which they are operated. The microbial detection method indirectly reflects the toxin content by detecting and analyzing the species, number, activity, and other characteristics of toxin-producing fungi. It is very intuitive, has low costs, and is primarily characterized by growth assays, microbial enumeration, the physiological index method, the molecular biology method, microscopy, and so on [20,21,22]. At present, there are few reports on the detection of vomitoxin by microbial detection methods, including a microbial detection method for Bacillus cereus vomitoxin, mainly through the PCR method of detecting the differentiation of toxin-producing CES genes to confirm the existence of toxin-producing strains of bacteria and through a quantitative analysis of the chemical method, which often requires complex pre-treatment processes and has a relative detection time [23].

Quantum dot nanospheres (QDs) are nanoscale spherical structures made of quantum dots (QDs), which have become a key research direction in nanotechnology and materials science in recent years. They have unique optical and electronic properties and are widely used in the fields of biomarkers, sensors, and optoelectronics, among other things [24]. The 2023 Nobel Prize in Chemistry was awarded to Moungi Bawendi, Louis Brus, and Alexei Ekimov for their contributions to the discovery and synthesis of quantum dots. Quantum dots exhibit high photoluminescence quantum yields and maintain stability under illumination. They resist photodegradation and feature phototunable properties [25]. This allows for precise control over luminescent colors by altering their chemical composition and size. Furthermore, as their size diminishes, both the absorption and emission peaks shift towards the blue end of the spectrum. In certain instances, they can absorb multiple low-energy photons to emit a single high-energy photon [26]. This characteristic broadens their potential applications in the realm of biomarkers. However, the small dimensions of quantum dots, typically ranging from 2 to 10 nm, reduce their biocompatibility and increase their tendency to agglomerate [27]. This significantly restricts the use of quantum dots in immunodetection applications. Relevant research indicates that silica exhibits excellent biocompatibility and low cytotoxicity. Its strong dispersion and stability properties, coupled with its low production costs and ease of preparation, make it an effective bio-carrier. Consequently, silica has been used in a wide array of applications in the biomedical field [28]. Zhang et al. [29] utilized the reverse microemulsion method to encapsulate single oleic-acid-coated CdSe/CdxZn1-xS core–shell quantum dots within SiO_2_ nanoparticles. This approach altered the hydrophobic characteristics of the quantum dots and enhanced their stability in aqueous solutions. Additionally, the silica encapsulation restricted the release of heavy metal ions, effectively converting the nonpolar-capped quantum dots into hydrophilic and biocompatible variants. YANG et al. [30] developed a dual-color SiO_2_@QD probe, achieving monodispersity and notable luminescence characteristics. They accomplished this by adsorbing red CdSe/ZnS-COOH (with an emission peak at 625 nm) and green CdSe/ZnS-COOH (with an emission peak at 525 nm) onto the SiO_2_ NPs’ surface through the classical adsorption mechanism involving poly (vinylidene) imide. This approach enabled the concurrent quantitative detection of C-reactive protein (CRP) and procalcitoninogen (PCT) in blood serum. By designing two lateral flow assay (LFA) systems, each with distinct luminescence signals, the limits of detection reached 0.5 ng/mL for CRP and 0.05 ng/mL for PCT. ZHANG et al. [31] successfully adsorbed cationic polyethyleneimine onto the surface of SiO_2_ NPs to create positively charged intercalated SiO_2_@PEI NPs. This method facilitated the intensive adsorption of quantum dots, resulting in the synthesis of composite nanomaterials, SiO_2_@PEI-QDs. These composites demonstrated superior dispersion, remarkable stability, and excellent luminescence properties. Furthermore, lateral flow assay (LFA) test strips derived from this probe exhibited a high sensitivity and accuracy for the rapid detection of Salmonella typhimurium in milk samples, achieving a lower limit of detection of 5 × 10^2^ CFU/mL.

Herein, SiO_2_ NPs were synthesized through the Stöber method. Subsequently, CdSe/ZnS QDs self-assembled onto the SiO_2_ NPs’ surface by leveraging the electrostatic adsorption of poly (vinylimine). Finally, the monoclonal antibody (mAb) for deoxynivalenol (DON) covalently attached to the carboxyl groups on the SiO_2_@QDs via the EDC/NHS chemical coupling method (Figure 1). This process resulted in a high-performance immumo-SiO_2_@QDs-DON mAb probe for the effective fluorescence immunodetection of DON.

## 2. Materials and Methods

### 2.1. Reagents and Materials

Carboxylated quantum dots (CdSe/ZnS) were purchased from Suzhou Xingshuo Nanotec Co., Ltd., Suzhou, China. N-hydroxythiosuccinimide (NHS), 1-ethyl (3-dimethylaminopropyl) carbodiimide (EDC), anhydrous ethanol, polyethyleneimine (PEI), tetraethyl silicate (TEOS), bovine serum albumin (BSA), and Tween-20 were purchased from Shanghai Macklin Biochemical Co., Ltd., Shanghai, China. Ammonia (28%) and morphine ethanesulfonic acid (MES) were purchased from Shanghai Aladdin Biochemical Technology Co., Ltd., Shanghai, China. Vomitoxin monoclonal antibody was purchased from Wuhan Chundu Biotechnology Co., Ltd., Wuhan, China.

The TEM images of SiO_2_ NPs, CdSe/ZnS QDs, SiO_2_@PEI NPs, and SiO_2_@QDs were obtained using JEM-2100F TEM and Tecnai G_2_ F_2_ (Thermo Ltd., Waltham, MA, USA) operating at 200 kV. Elemental mapping images were recorded by energy-dispersive X-ray spectroscopy (EDS) using a Thermo Fisher Scientific-Tecnai G_2_ F_20_ microscope equipped with an STEM unit. Ultraviolet–visible (UV-vis) absorption spectra were recorded using a Shimadzu UV-2600i UV-vis spectrophotometer (Shimadzu, Kyoto, Japan). The RF-6000 fluorescence spectrophotometer (Shimadzu, Japan) was used to record the fluorescence spectrum of QDs, SiO_2_ NPs, SiO_2_@PEI NPs, and SiO_2_@QDs. The zeta potentials of all the synthesized nanocomposites were measured using ZSU3200 Zeta Sizer (Malvern, Midlands, UK).

### 2.2. Preparation of SiO_2_@QDs

The monodispersed SiO_2_@QDs with a core–shell nanostructure were fabricated via the seed growth method facilitated by polyethylenimine. Initially, we synthesized monodisperse SiO_2_ NPs using Stöber’s method. Due to their robust stability, these SiO_2_ NPs serve effectively as a stable core for the probe. This innovation addresses the issue of quantum dots (QDs) being prone to aggregation and settling. Subsequently, SiO_2_ NPs underwent dispersion in a PEI aqueous solution (0.5%, *v*/*v*) via ultrasonic treatment. During this process, PEI rapidly self-assembled on the surface of the negatively charged SiO_2_ NPs through classical adsorption. Ultimately, SiO_2_@PEI NPs were retrieved through centrifugation. Finally, through ultrasonic action, carboxylated CdSe/ZnS quantum dots adhered to the SiO_2_@PEI NPs’ surface via electrostatic interactions, resulting in the synthesis of SiO_2_@QDs. Following centrifugal separation, the SiO_2_@QDs were preserved in an anhydrous ethanol solution.

### 2.3. Preparation of Immuno-SiO_2_@QDs

The immuno-SiO_2_@QD fluorescent probe was synthesized by conjugating anti-DON monoclonal antibodies onto the surface of SiO_2_@QDs by a carbodiimide-mediated chemical approach. In short, 1 mL of SiO_2_@QDs was centrifuged and dissolved in MEST (0.1 m) buffer mixed with 200 μL of EDC (5 mg/mL) and NHS (5 mg/mL) for ultrasound imaging for 15 min. EDC and NHS can form stable active esters with carboxyl groups on QDs’ surface, which provides conditions for the combination of anti-DON mAb in the future. EDC and NHS can form stable active esters with carboxyl groups on QDs’ surface, which can combine with amino groups on antibody proteins to form amide bonds. After mixed and centrifuged, carboxyl-activated SiO_2_@QDs and anti-DON mAb were incubated while being shaken in PBST buffer (0.05% Tween 20) at room temperature for 2 h. Then, 200 μL of 10% BSA was added to block the unreacted carboxyl sites of SiO_2_@QDs. Finally, the prepared immuno-SiO_2_@QDs were centrifuged and washed twice with PBST buffer (0.05% Tween 20). And the final precipitate was resuspended with 500 μL PBST buffer (0.05% Tween 20) and stored at 4 °C.

### 2.4. DON Detection in Grain Sample via Immuno-SiO_2_@QD Fluorescent Probe

We added a certain concentration of DON standard to non-toxic wheat samples and verified the reliability of our synthesized immune probe and detection method through the recovery rate of standard addition. At the constant temperature of 37 °C, 200 μL of immuno-SiO_2_@QDs was mixed with a series of DON standard solutions with gradient concentrations and incubated for 2 h. Then the standard curve was established with the fluorescence quenching value of immuno-SiO_2_@QDs and DON concentration as the coordinate axis. Due to the establishment of the standard curve, we can easily detect the spiked value of wheat samples by the above methods. The error between the added value and the actual added value can be used to judge the actual detection effect of immuno-SiO_2_@QDs.

## 3. Results and Discussion

### 3.1. Characterization of SiO_2_@QDs

The synthesized SiO_2_@QDs were characterized using transmission electron microscopy (TEM) and energy-dispersive spectroscopy (EDS). Figure 2a–d present the transmission electron microscope images of SiO_2_ nanoparticles (NPs), CdSe/ZnS quantum dots (QDs), SiO_2_@PEI NPs, and SiO_2_@QDs, respectively. Compared with QDs, SiO_2_ NPs have better dispersibility and a uniform particle size. After ultrasonic treatment for 40 min, a thin PEI interlayer was self-assembled on the surface of SiO_2_ NPs. Through the electrostatic adsorption effect of PEI, it can be seen from Figure 2d that carboxylated quantum dots are fixed on the surface of SiO_2_ NPs with good dispersibility and a uniform particle size. These quantum dots provide rich carboxyl sites and fluorescence signals for the subsequent antibody coupling.

Comprehensive elemental characterization was conducted through energy-dispersive X-ray spectroscopy (EDS) mapping as shown in Figure 2e–j. The elemental distribution profiles reveal significant co-localization of selenium (Se) and zinc (Zn) with silicon (Si) in the composite structure (Figure 2f). A quantitative EDS analysis (Figure 2k) confirms the predominant constituent elements as silicon (Si), cadmium (Cd), zinc (Zn), selenium (Se), and sulfur (S), with characteristic atomic percentages consistent with the designed SiO_2_@QDs’ architecture. Notably, the observed copper (Cu) signals are attributed to the copper grid substrate used for TEM sample preparation. The structure of the SiO_2_@QDs can also be verified from the side by a zeta potential analysis. As shown in Figure 2m, the initial potentials of the SiO_2_ NPs and CdSe/ZnS QDs are −48.1 and −14.35 mV, respectively. After a PEI interlayer is self-assembled on the surface of the SiO_2_ NPs, the potential of the SiO_2_@PEI NPs increases to 51.95 mV. After the adsorption of the CdSe/ZnS QDs, the potential finally dropped to 21.85 mV. From this change in potential, we can intuitively observe the synthesis stage of the SiO_2_@QD NPs and also prove that the formation of the SiO_2_@QDs is the result of the electrostatic adsorption effect mediated by PEI.

### 3.2. Characterization of Fluorescence Properties of SiO_2_@QDs

As shown in Figure 3, the fluorescence properties of the SiO_2_@QDs were characterized by the fluorescence emission properties and the fluorescence signal intensity of SiO_2_ NPs, CdSe/ZnS QDs, SiO_2_@PEI NPs, and SiO_2_@QDs under ultraviolet lamp irradiation, and the successful preparation of SiO_2_@QDs was verified. As shown in Figure 3a, under the excitation of a 365 nm ultraviolet lamp, both the CdSe/ZnS QDs and SiO_2_@QDs emitted strong red fluorescence, while the SiO_2_ NPs and SiO_2_@PEI NPs did not emit florescence, which proved that the fluorescence signal of the SiO_2_@QDs came from the quantum dots adsorbed on the surface.

Figure 3b and Figure 3c are the UV absorption spectra and PL emission spectra of nanoparticles, respectively. Through the UV absorption spectrum, it can be seen that the ultraviolet absorption peak of the CdSe/ZnS QDs is 284 nm, while the ultraviolet absorption peak of the SiO_2_@QDs is 294 nm, which may be due to the significant increase in the particle size of the quantum dots due to the combination of the SiO_2_ NPs, resulting in the red shift of the plasma resonance peak. In addition, because a large number of CdSe/ZnS QDs were connected around a single SiO_2_ NP at the same time, both the ultraviolet absorption intensity and the fluorescence emission intensity of the SiO_2_@QDs were greatly enhanced, which proves that taking SiO_2_ NPs as the core layer has a significant effect on improving the fluorescence properties of quantum dots and shows great potential for fluorescence immunoassays. At the same time, it can be seen from the PL emission spectrum that there were no fluorescence signals detected in the SiO_2_ NPs and SiO_2_@PEI NPs, which verifies the result that the above single UV light source has no fluorescence.

### 3.3. Stability Test of SiO_2_@QDs

In order to verify the stability of SiO_2_@QDs, the synthesized SiO_2_@QDs were suspended in ethanol solution and stored in the dark at 4 °C. The fluorescence intensity was measured every 10 days for 90 days, and the changes in fluorescence intensity were observed. It can be observed from Figure 4 that during the 90-day storage process, the fluorescence intensity of the SiO_2_@QDs showed a horizontal trend as a whole, and there was no obvious decrease, which proved that the stability of the SiO_2_@QDs was good.

### 3.4. Establishment of Immunodetection Method Based on SiO_2_@QD Fluorescent Probe

#### 3.4.1. Optimization of Antibody Labeling Quantity

In this method, the vomitoxin antibody is directly coupled with the carboxyl group on the surface of SiO_2_@QDs by the carbodiimide chemical method. In this way, as shown in Figure 5, carbodiimide (EDC) reacts with carboxyl to form a stable intermediate active ester, and N-hydroxysuccinimide (NHS) reacts with this active ester to form a more stable NHS ester. This step can provide a lower reaction selectivity and fewer by-products, and at the same time can reduce the damage to antibodies. Then, through a covalent connection, the reaction product reacts with the amino group on the antibody molecule, and the antibody molecule is coupled to the surface of the SiO_2_@QDs, so as to prepare the immuno−SiO_2_@QDs−DON mAb probe. In this process, the potential of the SiO_2_@QDs changes because the carboxyl group reacts with the amino group to form an amide bond. Therefore, the successful coupling of the DON antibody with the SiO_2_@QDs can be verified by the change in zeta potential, and of course, the best antibody labeling amount can also be selected by the change in potential.

As shown in Figure 6, with the increase in the amount of DON mAb antibody, the zeta potential gradually decreased. When the amount of antibody added was 15 μg, the zeta potential no longer changed, which proved that the DON mAb antibody was successfully coupled with SiO_2_@QDs and reached the maximum saturation. At the same time, in order to avoid the excessive use of antibodies, 15 μg was selected as the best dosage of the labeled antibody.

#### 3.4.2. Performance Test of Immuno-SiO_2_@QDs-DON mAb Fluorescent Immunoprobe

The SiO_2_@QDs-DON mAb probe is at an initial fluorescence intensity when it is not bound to the target, and when it is bound to the target, it leads to fluorescence resonance energy transfer (FRET) between the SiO_2_@QD-DON mAb probe and the target molecule, which dissipates the excited state energy of the SiO_2_@QD-DON mAb probe in a non-radiative form, resulting in a decrease in fluorescence intensity, i.e., the fluorescence burst phenomenon [32]. Based on this principle, we set a concentration gradient of 0, 2, 4, 8, 16, 32, 64, 128, and 256 ng/mL standards to perform fluorescence immunoreactivity tests with 200 μL of the SiO_2_@QDs-DON mAb probe and detected the post-immunization fluorescence intensity values using a fluorescence spectrophotometer. Then, the standard working curve is drawn with the logarithmic value of the DON concentration as the abscissa and the fluorescence intensity quenching value before and after immunization as the ordinate. As shown in Figure 7, there was a good linear relationship between fluorescence intensity quenching value and the logarithmic value of DON concentration in the range of 0–16 ng/mL. The regression equation shown is “ΔF = 6836.36x − 1492.46”, R2 = 0.999 (“ΔF” is the fluorescence intensity quenching value before and after immunization and “x” is the logarithmic value of the DON concentration), and the detection limit is 0.25 ng/mL. Table 1 is a part of the relevant research on the instrument detection methods of emetic toxins in grains at home and abroad. The comparison shows that the immuno-SiO_2_@QD fluorescent probe has a high sensitivity in the immunoassay of vomitoxin compared with the physicochemical method. This avoids having to perform a cumbersome operation procedure and rely on large-scale precision instruments, as in the physicochemical analysis method.

#### 3.4.3. Validation of Specific Performance of Immune-SiO_2_@QD Fluorescent Probes

To verify the specificity of the immuno-SiO_2_@QD fluorescent probe, we used the above method to detect several mycotoxins structurally similar to the vomitoxin (DON), such as aflatoxin B1 (AFB1), aflatoxin B2 (AFB2), T-2 toxin, hetrotoxin (OTA), and zearalenone (ZEN). As demonstrated in Figure 8, except for vomitoxin (DON), the immuno-SiO_2_@QD fluorescent probe barely recognized the other mycotoxins.

#### 3.4.4. Determination of Recovery Rate of Vomitoxin in Actual Wheat Flour Samples by Standard Addition

In order to evaluate the performance of the immuno-SiO_2_@QDs-DON mAb fluorescent probe for actual sample detection, we added the DON standard solution with concentrations of 5, 10, and 15 ng/mL to the wheat flour samples, which were detected to be non-toxic, repeated the detection process three times for each spiked sample, carried out the fluorescence detection test, and calculated the recovery rate. As shown in Table 2, for the wheat flour samples spiked with these concentrations of the standards, the detection of vomitoxin ranged from 92.2 to 101.6%, and the relative standard deviation is 3.4–4.6%, which confirms the reliability of the detection results. The immuno-SiO_2_@QD fluorescent probe can be used to detect real wheat flour samples.

## 4. Conclusions

The synthesis of the immuno-SiO_2_@QDs-DON mAb probe aims to gather CdSe/ZnS QDs (quantum dots) on the surface of SiO_2_ NP microspheres by the electrostatic adsorption effect of PEI. This method effectively mitigates the easy agglomeration of quantum dots and improves their stability. At the same time, the fluorescence intensity of the probe is improved because of the combination of a large number of quantum dots on the surface of a single SiO_2_ NP microsphere, and the probe can be stored stably for a long time at 4 °C. The EDC/NHS chemical coupling method makes the anti-DON monoclonal antibody firmly bind to the carboxyl groups on the surface of quantum dots in a covalent way. The immunofluorescence detection method based on an immuno-SiO_2_@QDs-DON mAb probe was proven to have a high sensitivity, as well as excellent reproducibility and stability. In operation, it only needs to be incubated for 2 h at 37 °C, which avoids the complicated process of repeated washing combined with the traditional ELISA method, is simple to operate, is capable of rapid detection, and shows great application potential for the instant detection of grain vomitoxin.

## Figures and Tables

**Figure 1 foods-14-01545-f001:**
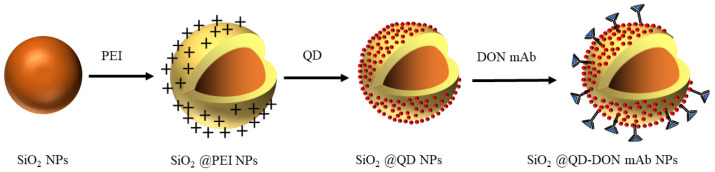
Schematic flow diagram of SiO_2_@QD fluorescent microsphere probe synthesis.

**Figure 2 foods-14-01545-f002:**
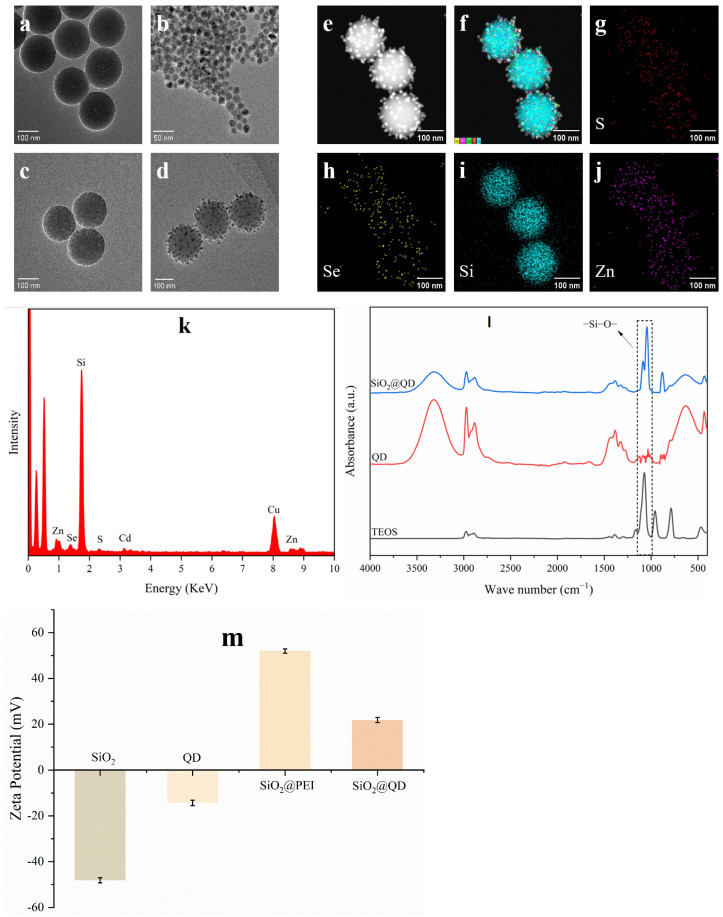
Characterization of the SiO_2_@QDs. (**a**–**d**) TEM images of SiO_2_ NPs, CdSe/ZnS QDs, SiO_2_@PEI NPs, and SiO_2_@QDs. (**e**–**j**) EDS elemental mapping images showing the elemental distributions of S (red), Se (yellow), Si (blue), and Zn (purple) in the SiO_2_@QD NPs. (**k**) EDS spectrum of a single SiO_2_@QD NPs. The Cu peak is from the Cu TEM grid. (**l**) FT-IR of SiO_2_@QDs, CdSe/ZnS QDs, and TEOS. Compared with QDs, TEOS and SiO_2_@QDs have obvious changes in the range of 1000–1300 cm^−1^, which is caused by the stretching vibration of Si-O groups. (**m**) Zeta potentials of QDs, SiO_2_ NPs, SiO_2_@PEI NPs, and SiO_2_@QD NPs.

**Figure 3 foods-14-01545-f003:**
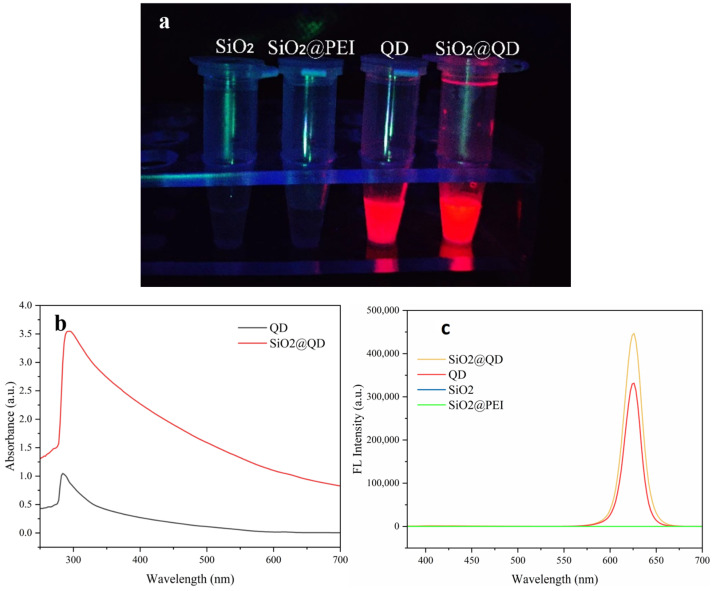
Characterization of fluorescence properties of SiO_2_@QDs. Photographs (**a**), ultraviolet spectra (**b**), and fluorescence spectra (**c**) of SiO_2_ NPs, SiO_2_@PEI NPs, and SiO_2_@QDs under 365 nm UV light.

**Figure 4 foods-14-01545-f004:**
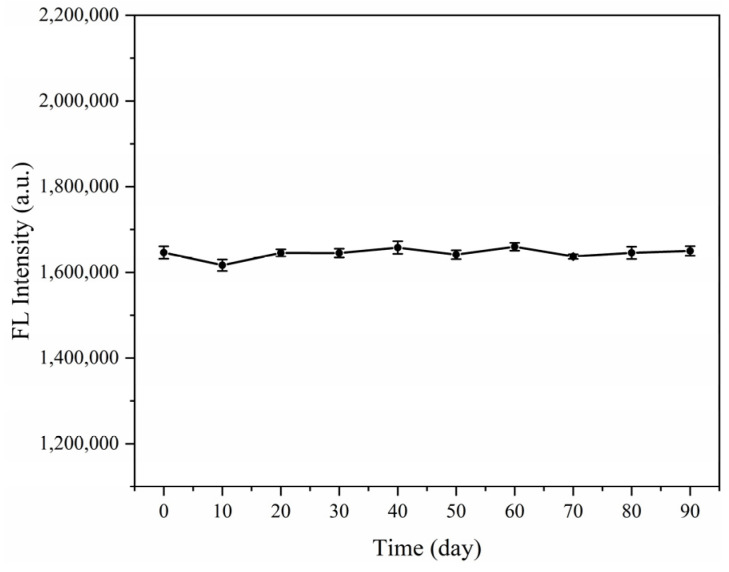
The fluorescence intensity stability of SiO_2_@QDs.

**Figure 5 foods-14-01545-f005:**
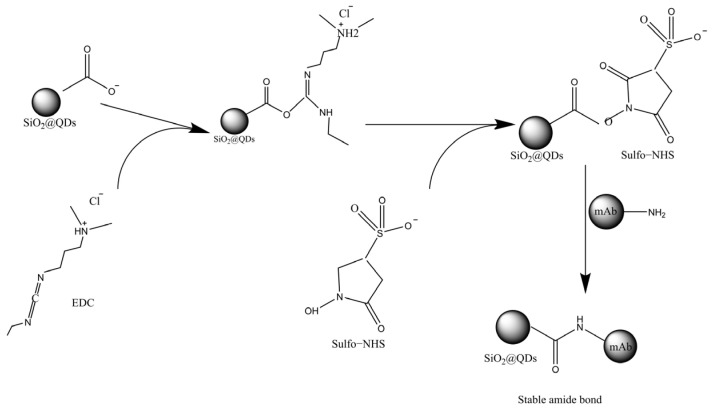
Principle of EDC/NHS chemical coupling method.

**Figure 6 foods-14-01545-f006:**
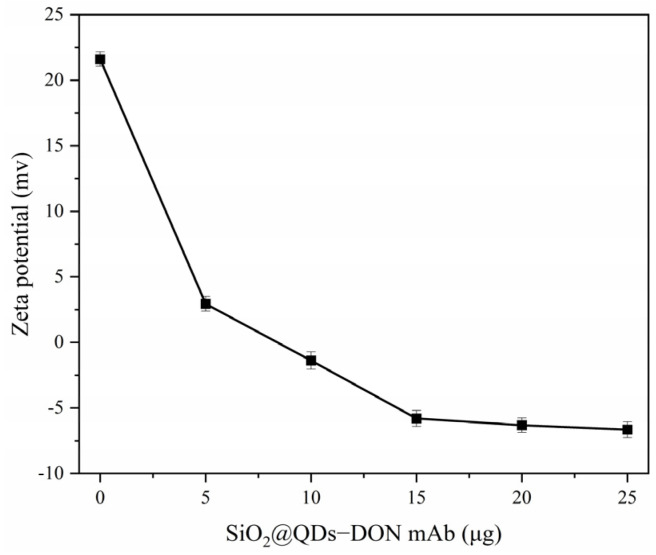
Optimization of DON mAb labeling amount.

**Figure 7 foods-14-01545-f007:**
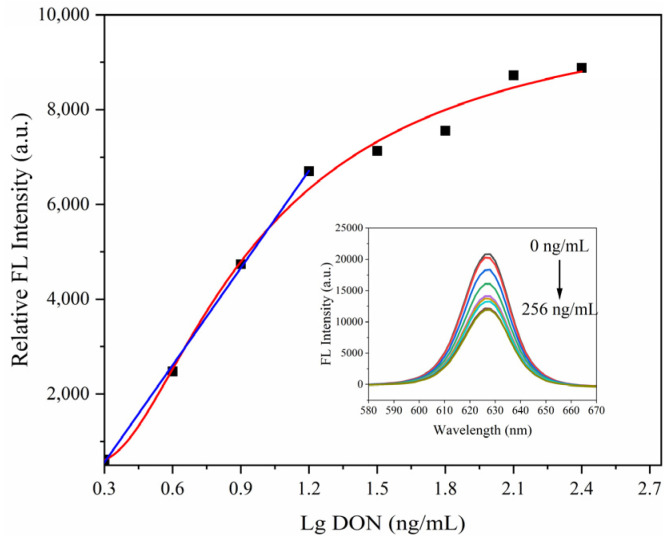
Working curve of DON fluorescence immunoassay.

**Figure 8 foods-14-01545-f008:**
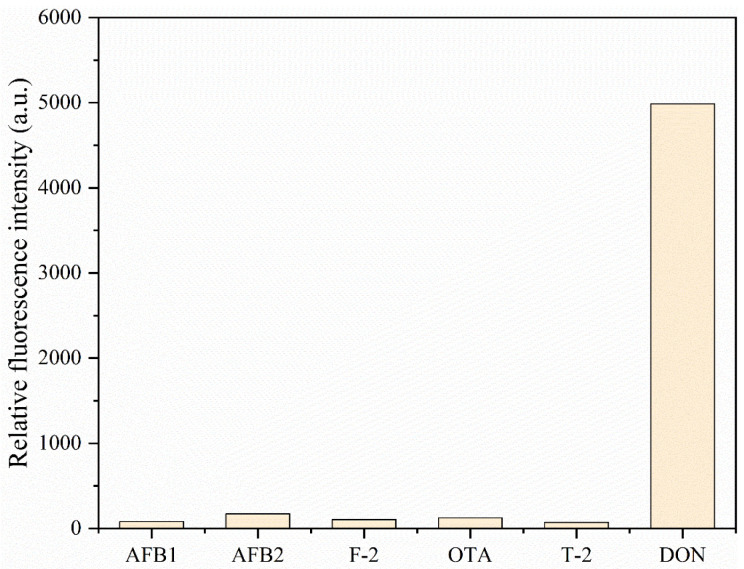
Immuno-SiO_2_@QDs-DON mAb fluorescent probe specificity assay.

**Table 1 foods-14-01545-t001:** Comparison of detection limits of different detection methods for vomitoxin.

Analyte	Detection Method	Detection Limit	Reference
Wheat flour	HPLC	48 ng/mL	[33]
Wheat flour	LC-MS/MS	4 μg/kg	[34]
Wheat flour	SPE-LC-MS/MS	0.1–0.2 μg/kg	[35]
Corn and oats	LC-MS/MS	0.04 μg/kg	[36]
Rice and bran	SPE-HPLC-UV	<11.9 μg/kg	[37]

**Table 2 foods-14-01545-t002:** The recovery rate of DON in wheat flour samples.

Samples	Spiked (ng/mL)	Test (ng/mL)	Recoveries (%)	RSD (%)
wheat flour	5	4.61	92.2	3.6
	10	10.16	101.6	4.6
	15	14.41	96.1	3.4

## Data Availability

The original contributions presented in this study are included in the article. Further inquiries can be directed to the corresponding author.

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
