# Peer review of "A Highly Sensitive Silicon-Core Quantum Dot Fluorescent Probe for Vomitoxin Detection in Cereals"

_foods, 2025, doi:10.3390/foods14091545_

Round 1
Reviewer 1 Report
Comments and Suggestions for Authors
MS entitled “A Highly Sensitive in Silico Quantum Dot Fluorescent Probe for the Detection of Vomitoxin in Cereals” by Dong et al. 2025 has been reviewed. This topic is very important for the development of rapid methods for the detection of toxic fungal metabolites (like DON) in cereals. I suggest major revision. Some major comments are given below.
The material and methods are not clearly presented. Please add for each method how is protocol conducted including sample preparation and other. Did you perform a statistical test? Which method author used for validation their obtained results? Lines 270: “to the wheat flour samples which were detected to be non-toxic”– what that means? and which method author used to know that level DON is below permitted value.
Author Response
|
Comments 1: The material and methods are not clearly presented. Please add for each method how is protocol conducted including sample preparation and other. |
|
Response 1: I Agree. We have added and shown in red font lines 150 to 188 of the article. |
|
Comments 2: Did you perform a statistical test? Which method author used for validation their obtained results? |
|
Response 2: Thank you for pointing this out. We did not perform statistical tests because, by reviewing a large amount of relevant literature, we found very few articles with relevant detection methods that applied statistical tests. Instead, we designed several experiments and calculated the relative standard deviation (RSD) of the spiked samples to reflect the accuracy and reliability of the assay, which was also applied in a large number of articles of the same type. For example Chen Y, Liu X, Li J, et al. Development of a Sensitive Enzyme Immunoassay Using Phage-Displayed Antigen-Binding Fragments for Zearalenone Detection in Cereal Samples. foods. 2025;14(5):746. Published 2025 Feb 22. doi:10.3390/foods14050746. At the same time, because of the addition of a known amount of vomitoxin standard to the negative samples, we detected such spiked samples by the developed assay , and the results were validated by observing the difference between the detected amount and the spiked amount.
|
|
Comments 3: Lines 270: “to the wheat flour samples which were detected to be non-toxic”– what that means? |
|
Response 3: Thank you for pointing this out. The meaning of this statement is that this sample has been tested and proven to be free of vomitoxin. The purpose of this test is also that the amount of vomitoxin standard that we add to the wheat flour sample is the vomitoxin toxin level of the wheat flour, which makes it easier for us to validate the accuracy of the test method. |
|
Comments 4: which method author used to know that level DON is below permitted value. |
|
Response 4: Thank you for pointing this out. By summarizing the maximum limits of vomitoxin in the relevant standards and regulations in lines 52 to 55 of the introduction: GB 2761-2017 “National Standard for Food Safety: Limit of Mycotoxins in Foods” stipulates that the maximum limit of DON in cereals and cereal products should not exceed 1000 μg/kg, and the European Union stipulates that the upper limit of vomitoxin in baby food is 200 μg/kg, it can be known that the level of vomitoxin recognized by this method is much lower than the maximum limit required by the relevant standards. It can be seen that the amount of vomitoxin recognized by this method is much lower than the maximum limit required by the relevant standards. Meanwhile, the comparison in Table 1, line 312 of the article also shows that this method is superior to the physical and chemical detection methods. |

Reviewer 2 Report
Comments and Suggestions for Authors
In “A Highly Sensitive in Silico Quantum Dot Fluorescent Probe for the Detection of Vomitoxin in Cereals,” the authors present the preparation of an immuno-silica@QDs fluorescent probe and its application for the detection of vomitoxin (DON). The characterization of the materials is appropriate, and the results are promising, as the method used for the determination of DON is extremely simple and fairly sensitive.
However, the main issue is that the authors do not provide an adequate explanation of how the detection of DON actually works. Essentially, the fluorescent probe is mixed with the sample, the antibodies bind to DON, and fluorescence decreases. The cause of this quenching is not clearly explained—especially considering that DON is a small molecule. The authors can provide solid arguments to support the underlying mechanism of their analytical method, pending a thorough revision of the language.
The title of the paper must be changed, as “in silico” refers to a computational approach rather than an experimental one.
The meanings of PEI and DON should be defined in the abstract.
In line 40, the authors should specify the country they are referring to.
The caption of Figure 1—“Fluorescence immunoassay of deoxynivalenol based on SiOâ‚‚@QDs microsphere probe”—does not accurately describe the figure’s content and should be revised accordingly.
The Materials and Methods section should include detailed experimental information on how DON detection was carried out, including incubation times, sample and reagent volumes and concentrations, and the procedure used to spike the wheat samples.
In lines 246–247, the authors state: “there is a good linear relationship between fluorescence intensity quenching value and DON concentration in the range of 0–16 ng/mL.” However, the actual relationship is between fluorescence intensity and the logarithm of the DON concentration.
There are numerous language errors that require correction—for example, in lines 57, 150, and 158. Additionally, the expression “at Pique level” (line 254) is unclear; its meaning should be clarified or replaced. The term “average recovery rate” (line 272) is also misleading, as this is not a rate (i.e., a change over time), but simply a percentage of recovery.
Author Response
|
Comments 1: However, the main issue is that the authors do not provide an adequate explanation of how the detection of DON actually works. |
|
Response 1: We are also aware of this problem. In the article, we have verified the detection performance of the prepared probe by adding a certain amount of standard to a non-toxic wheat flour sample and then performing the assay. We use this method as an evaluation of the practical application performance of the probe. Through the review of related literature, we learned that this approach is also effective. As for the testing of real commercially available samples, in view of the time and financial requirements, we were unable to complete the collection of commercially available samples and verify the real value of the toxin content of the collected samples by liquid phase and other relevant instrumental methods. We will focus on this aspect of the study in the subsequent sessions. |
|
Comments 2: Essentially, the fluorescent probe is mixed with the sample, the antibodies bind to DON, and fluorescence decreases. This cause of this quenching is not clearly explained—especially considering that DON is a small molecule. The authors can provide solid arguments to support the underlying mechanism of their analytical method, pending a thorough revision of the language. |
|
Response 2: Agree. I have explained the reason for the binding of the probe to the toxin leading to the phenomenon of fluorescence bursting in lines 289 through 294 of the article and shown it in red font, and added the literature support for this phenomenon in lines 441 and 442 of the article and shown it in red font. |
|
Comments 3: The title of the paper must be changed, as “in silico” refers to a computational approach rather than an experimental one. |
|
Response 3: Agree. Since the probes I synthesized were in silico quantum dot probes, “in silico” in my original meaning is silicon-based. This may have caused a misunderstanding, so I put a “-” sign between “Silico” and “Core” in line 2 and italicized the proper noun. |
|
Comments 4: The meanings of PEI and DON should be defined in the abstract. |
|
Response 4: Agree. In view of the need to fine-tune the abstract, we have summarized the meaning of “vomitoxin” in lines 9, 10 and 11 of the article and shown it in red font. The term “polyethyleneimine” is summarized in lines 13, 14, 15 and 17 of the article and shown it in red font. |
|
Comments 5: In line 40, the authors should specify the country they are referring to. |
|
Response 5: I Agree. We've made the change in line 44 of the article and shown it in red font. |
|
Comments 6: The caption of Figure 1—“Fluorescence immunoassay of deoxynivalenol based on SiOâ‚‚@QDs microsphere probe”—does not accurately describe this figure’s content and should be revised accordingly. |
|
Response 6: Agree. We've made the change in line 131 of the article and shown it in red font. |
|
Comments 7: The Materials and Methods section should include detailed experimental information on how DON detection was carried out, including incubation times, sample and reagent volumes and concentrations, and the procedure used to spike this wheat samples. |
|
Response 7: Agree. We have added and shown in red font lines 150 to 188 of the article. |
|
Comments 8: In lines 246–247, the authors state: “there is a good linear relationship between fluorescence intensity quenching value and DON concentration in the range of 0–16 ng/mL.” However, the actual relationship is between fluorescence intensity and the logarithm of the DON concentration. |
|
Response 8: Agree. We've made the changes and shown them in red on lines 300 and 301 of the article. |
|
Comments 9: There are numerous language errors that require correction—for example, in lines 57, 150, and 158. Additionally, the expression “at Pique level” (line 254) is unclear; its meaning should be clarified or replaced. The term “average recovery rate” (line 272) is also misleading, as this is not a rate (i.e., a change over time), but simply a percentage of recovery. |
|
Response 9: Agree. Firstly, regarding the language errors in the article, we have made changes in lines 58 to 62, 192 to 195 and 202 to 209 of the article and shown them in red. Secondly, with regard to the ambiguity of the expression “at Pique level”, we have corrected it in lines 306 to 309 of the article and shown it in red. Thirdly, with regard to the misleading reference to “average recovery rate”, this has been changed in lines 377 and 378 of the article and shown in red. |
